# Targeting Macrophages: Therapeutic Approaches in Diabetic Kidney Disease

**DOI:** 10.3390/ijms25084350

**Published:** 2024-04-15

**Authors:** Da-Wei Lin, Tsung-Ming Yang, Cheng Ho, Ya-Hsueh Shih, Chun-Liang Lin, Yung-Chien Hsu

**Affiliations:** 1Department of Internal Medicine, St. Martin De Porres Hospital, Chiayi City 60069, Taiwan; orcaking88@gmail.com; 2Division of Pulmonary and Critical Care Medicine, Chang Gung Memorial Hospital, Chiayi County 61363, Taiwan; n120633@cgmh.org.tw; 3School of Traditional Chinese Medicine, College of Medicine, Chang Gung University, Taoyuan 33303, Taiwan; linchunliang@cgmh.org.tw; 4Division of Endocrinology and Metabolism, Chang Gung Memorial Hospital, Chiayi County 61363, Taiwan; hc1238@cgmh.org.tw; 5Departments of Nephrology, Chang Gung Memorial Hospital, Chiayi County 61363, Taiwan; rita1608@gmail.com; 6Kidney and Diabetic Complications Research Team (KDCRT), Chang Gung Memorial Hospital, Chiayi County 61363, Taiwan; 7Kidney Research Center, Chang Gung Memorial Hospital, Taipei 10507, Taiwan; 8Center for Shockwave Medicine and Tissue Engineering, Chang Gung Memorial Hospital, Kaohsiung 83301, Taiwan; 9School of Medicine, College of Medicine, Chang Gung University, Taoyuan 33303, Taiwan

**Keywords:** diabetic kidney disease, macrophage, polarization

## Abstract

Diabetes is not solely a metabolic disorder but also involves inflammatory processes. The immune response it incites is a primary contributor to damage in target organs. Research indicates that during the initial phases of diabetic nephropathy, macrophages infiltrate the kidneys alongside lymphocytes, initiating a cascade of inflammatory reactions. The interplay between macrophages and other renal cells is pivotal in the advancement of kidney disease within a hyperglycemic milieu. While M1 macrophages react to the inflammatory stimuli induced by elevated glucose levels early in the disease progression, their subsequent transition to M2 macrophages, which possess anti-inflammatory and tissue repair properties, also contributes to fibrosis in the later stages of nephropathy by transforming into myofibroblasts. Comprehending the diverse functions of macrophages in diabetic kidney disease and regulating their activity could offer therapeutic benefits for managing this condition.

## 1. Introduction

Chronic kidney disease (CKD) represents a notable worldwide health issue that impacts more than 10% of the global population [1]. It is recognized for its considerable adverse effects on both quality of life and mortality rates [2]. Diabetes mellitus represents a significant risk factor for chronic kidney disease (CKD), especially in developed countries and Asia [3,4]. In the United States, statistics from the Chronic Kidney Disease Surveillance System of the Centers for Disease Control and Prevention reveal that 14% of individuals aged 20 and above are affected by chronic kidney disease (CKD), and among them, 30% also have diabetes (http://www.cdc.gov/ckd accessed on 18 March 2024).

The hyperglycemic conditions present in diabetes play a significant role in the generation of glucose degradation products and advanced glycation end products. Furthermore, it exacerbates the renin–angiotensin–aldosterone system, resulting in hemodynamic alterations in the glomeruli and the initiation of diabetic kidney disease via inflammatory pathways. The inflammatory response entails the infiltration of immune cells, notably macrophages, whose involvement in the advancement of diabetic kidney disease has been emphasized in several studies. The presence of macrophages in the kidneys has been associated with the progression of diabetic nephropathy in *db/db* mice. This condition is distinguished by persistent hyperglycemia, deposition of immune complexes in the glomeruli, and heightened production of chemokines in the kidneys [5]. In vitro investigations have demonstrated that heightened glucose levels or advanced glycation end products (AGE) have the potential to stimulate macrophages. This stimulation can result in heightened expression of intracellular adhesion molecule-1 (ICAM-1) on tubular cells and the secretion of active transforming growth factor-β1 (TGF-β1) by macrophages [6]. An analysis of renal autopsy specimens from 88 individuals with type 2 diabetes and confirmed diabetic nephropathy demonstrated a significant association between the presence of glomerular CD163+ macrophages and the extent of diabetic nephropathy, interstitial fibrosis, tubular atrophy, and glomerulosclerosis. The presence of interstitial CD68+ macrophages exhibited an inverse relationship with the glomerular filtration rate and a positive correlation with albuminuria [7]. Recent developments in the analysis of single-cell RNA sequencing (scRNA-seq) have revealed a rise in immune cell populations within the diabetic glomeruli of streptozotocin-induced diabetic mice. Cluster analysis has shown a predominance of macrophages in this context [8]. A single-cell transcriptomic analysis of CD45-enriched kidney immune cells from OVE26 mice with type 1 diabetes revealed an increase in macrophage subsets over time, concomitant with elevated pro-inflammatory gene expression [9].

An in-depth analysis of research on the participation of macrophages in diabetic nephropathy can provide valuable insights into their varied functions in the advancement of this disease. This comprehension establishes the groundwork for regarding macrophages as plausible therapeutic targets for the management of diabetic nephropathy.

## 2. Activation of Macrophages and Recruitment of Monocytes in Early Stage of Diabetic Kidney Disease

During the early phases of hyperglycemia, macrophages were attracted to the glomeruli, and their appearance was associated with increased levels of α1-chain type IV collagen messenger ribonucleic acid (mRNA). This recruitment was preceded or accompanied by an upregulation in the expression of vascular cell adhesion molecule 1 (VCAM-1), ICAM-1, and monocyte chemoattractant protein 1/Chemokine (C-C motif) ligand 2 (MCP-1/CCL2), which promote monocyte migration. Moreover, the synthesis of interleukin-1β mRNA in the glomerulus was increased in the initial stages, potentially playing a role in the upregulation of extracellular matrix (ECM) and adhesion molecule gene expressions [10].

In addition to adhesion molecules, the C-C chemokine receptor (CCR)2 is essential in the regulation of monocyte/macrophage migration into injured tissues. The administration of a selective CCR2 antagonist to diabetic mice over a 12-week period led to notable reductions in albuminuria, blood urea nitrogen, plasma creatinine levels, histological alterations, and glomerular macrophage recruitment in comparison to the control group. Similarly, mice deficient in CCR2 (*CCR2^−/−^*) demonstrated a decrease in albuminuria, a reduction in fibronectin mRNA expression, and lower production of inflammatory cytokines compared to *CCR2*^+/+^ mice, despite having similar blood glucose levels [11,12].

AGEs have been shown to markedly elevate intracellular reactive oxygen species (ROS) generation, expression of inducible nitric oxide synthase (iNOS), levels of receptor for advanced glycation end products (RAGE) and toll-like receptor-4 (TLR-4), as well as signal transducer and activator of transcription 1 (STAT1) expression. Additionally, AGEs have been found to enhance the phosphorylation levels of cytoplasmic STAT1 in infiltrated macrophages. Inhibition of the RAGE and suppression of TLR-4 resulted in reduced levels of phosphorylated STAT1, nuclear STAT1, and secretion of pro-inflammatory cytokines in macrophages exposed to AGEs. This suggests the involvement of the RAGE/ROS/TLR-4/STAT1 signaling pathway in promoting a pro-inflammatory phenotype in infiltrated macrophages [13].

The expression of the receptor for RAGE on immune cells originating from the hematopoietic system, in addition to macrophages, plays a role in the functional alterations observed in diabetic nephropathy. This involvement is manifested through the promotion of macrophage infiltration and the induction of renal tubulointerstitial damage. Mice with diabetes and wild-type characteristics that were reconstituted with bone marrow deficient in RAGE showed a decrease in urinary albumin excretion, podocyte loss, enhanced creatinine clearance, and reduced tubulointerstitial injury and fibrosis. The mice exhibited reduced levels of infiltrating CD68(+) macrophages with an activated phenotype and decreased expression of MCP-1/CCL2, macrophage migration inhibitory factor (MIF), and interleukin (IL)-6 in the renal cortex [14].

The systemic renin–angiotensin system (RAS) plays a crucial role in regulating extracellular fluid volume and arterial pressure, while a local RAS operates at the tissue level and contributes to non-hemodynamic functions [15]. In individuals with diabetes mellitus, there may be a paradoxical decrease in systemic RAS activity, potentially indicating activation of the intrarenal RAS [16]. Within the kidney, intrarenal Angiotensin II (Ang II) positively influences proximal tubular angiotensinogen, collecting duct renin, and tubular angiotensin II type 1 (AT1) receptors. Apart from the conventional RAS pathways, local Ang II production in the kidney involves pro-renin receptors and chymase. AT1 receptor-dependent mechanisms facilitate the active uptake of circulating Ang II into proximal tubular cells, resulting in higher Ang II concentrations in the renal interstitial fluid and proximal tubular compartments compared to the bloodstream [15]. Monocyte/macrophage can express ATR, angiotensinogen, renin, and angiotensin peptide hormone [17,18,19,20]. The recruitment of monocytes in diabetes-related kidney disease can be influenced by the RAS through upregulation of MCP-1, osteopontin, and various adhesion molecules including selectins, ICAM, and VCAM [21,22,23,24,25]. Activation of paracrine or endocrine signaling processes of RAS modulates the differentiation and polarization of monocytes. However, limited studies have explored the effects of Ang II on macrophage polarization in kidney disease. Drawing insights from in vitro studies, an atherosclerosis model, a deoxycorticosterone acetate (DOCA)/salt hypertension model, and the AMADEO clinical trial, it is proposed that targeting the RAS system could potentially condition macrophages to be less responsive to a harmful proinflammatory renal environment while retaining their ability to transition into the renoprotective anti-inflammatory phenotype [26,27,28,29,30]. Interestingly, the use of angiotensin-converting enzyme (ACE) inhibitor (e.g., enalapril) in experimental models of diabetic nephropathy demonstrates a temporary improvement in albuminuria. This improvement is paradoxically linked to the re-establishment of T cells and pro-inflammatory M1-like macrophages within the renal tissue [31]. The identification of aldosterone escape phenomenon in the context of sustained ACE inhibitor treatment, along with the observed impact of eplerenone in conjunction with ACE inhibitors in attenuating the advancement of diabetic nephropathy in diabetic mice, suggests a potential involvement of aldosterone in influencing the polarization of macrophages [32,33].

In a hyperglycemic setting, dysregulated metabolic elements could potentially enhance the activation of macrophages, thereby worsening inflammation and nephropathy. Disrupted metabolism of branched-chain amino acids (BCAAs) is concurrently observed in individuals with type 2 diabetes. Impaired BCAA catabolism is intricately linked to chronic inflammation and tissue damage in type 2 diabetes. This association is partially attributed to macrophage oxidative stress triggered by branched-chain ketoacids (BCKA). Prolonged supplementation of BCAA exacerbates levels of BCKA, inflammation, tissue fibrosis, and macrophage hyperactivation in *db/db* mice. Conversely, enhancing BCAA catabolism with a pharmacological activator alleviates these effects. The activation of BCKA is associated with elevated mitochondrial oxidative stress and redox imbalance in macrophages and diabetic tissues, highlighting the complex interaction between metabolic abnormalities and inflammation in the progression of diabetic nephropathy (DN) [34].

## 3. Crosstalk between Macrophages and Non-Myeloid Cells within the Renal Tissue of Diabetic Individuals 

In a Transwell co-culture model, it was noted that macrophages did not induce apoptosis in podocytes in the absence of elevated glucose levels. However, exposure to high glucose concentrations led to an increase in podocyte apoptosis. Co-culturing podocytes with macrophages under high-glucose conditions intensified this effect compared to podocytes cultured alone under the same conditions. The heightened glucose levels activated macrophages, prompting a transition towards a pro-inflammatory phenotype. Conditioned media from these pro-inflammatory macrophages, activated by high glucose, triggered apoptosis in podocytes [35]. (To efficiently locate information in this section, readers are encouraged to refer to Figure 1.)

A hyperglycemic milieu can impact both immune cells and non-myeloid cells within the renal system. Elevated glucose levels and associated lipotoxicity can initiate a stress response in podocytes, resulting in the generation of intracellular ROS, nuclear deoxyribonucleic acid (DNA) damage, alterations in gene expression, and abnormal nephrin expression. These modifications can lead to phenotypic changes, detachment, and apoptosis, ultimately disrupting the slit diaphragm and causing proteinuria [36]. Furthermore, this non-myeloid cell response to high glucose can interact with macrophages and monocytes, thereby amplifying the overall impact. Subsequent sections delve into the diverse interactions between macrophages and non-myeloid cells in diabetic nephropathy.

### 3.1. Crosstalk between Macrophage and Podocyte within Diabetic Kidneys

In humans, there is a well-established correlation between the activity of the growth hormone (GH)/insulin-like growth factor 1 (IGF-1) axis and renal conditions such as hypertrophy, microalbuminuria, and glomerulosclerosis, which mirror the development of diabetic nephropathy. Conversely, studies have shown that a deficiency in growth hormone (GH) or the inhibition of GH receptor (GHR) activity can offer protection against diabetic nephropathy [37]. A scRNA-seq analysis has revealed an increase in tumor necrosis factor-α (TNF-α) signaling in human podocytes after exposure to growth hormone. The media from these treated podocytes has been found to promote the differentiation of monocytes into macrophages. However, when this media was neutralized with a TNF-α antibody, the impact on monocyte differentiation was reduced. Treatment of mice with growth hormone resulted in heightened recruitment of macrophages, podocyte damage, and proteinuria [38]. Other factors like angiotensin II (ANG II), which is elevated in diabetic nephropathy, have also been implicated. In an in vivo experiment, podocytes pre-treated with ANG II exhibited increased TNF-α mRNA and protein expression in response to AGE [39]. Moreover, elevated levels of glucose and advanced glycation end products (AGE) trigger the upregulation of TLR-4 and MCP-1/CLL2 in podocytes by activating the RAGE through a nuclear factor κ-light-chain-enhancer of activated B cells (NF-κB)-dependent signaling pathway [40,41]. Podocytes cultured in high-glucose conditions showed significantly enhanced macrophage migration compared to those in normal glucose conditions [42]. Consequently, podocytes in a diabetic environment can release TNF-α and MCP-1, which promote macrophage migration and induce the expression of T-cell immunoglobulin and mucin domain-3 (TIM-3) on renal macrophages through the NF-κB/TNF-α pathway, thereby contributing to aggravate podocyte injury in diabetic kidney disease and exacerbating the progression of diabetic nephropathy [43].

### 3.2. Crosstalk of Macrophage and Epithelial Cell/Tubular Cell within Diabetic Kidney

Epsin 1, a member of a highly conserved family of membrane-associated, ubiquitin-binding endocytic adapter proteins, acts as a specific sorting protein for polyubiquitin in association with clathrin [44]. In individuals with diabetic nephropathy and *db/db* mice, there is an upregulation of Epsin1 and exosomal delta-like ligand 4 (DLL4) expression from tubular epithelial cells (TECs). This, along with the presence of the neurogenic locus notch homolog protein 1 (NOTCH1) intracellular domain (N1ICD) in kidney tissues, leads to tubulointerstitial damage. The activation of NOTCH signaling by DLL4 ligand promotes pro-inflammatory macrophage differentiation, inhibits anti-inflammatory macrophage polarization, and induces apoptosis by downregulating anti-inflammatory-specific gene expression [45]. Exosomes from tubular cells (HK-2) treated with high glucose and Epsin1 knockdown reduce macrophage activation, TNF-α, and IL-6 expression, and tubulointerstitial damage in C57BL/6 mice in vivo. In vitro studies have found DLL4-enriched exosomes in HK-2 cells exposed to high glucose. These exosomes are internalized by monocytes (THP-1 cells) and trigger pro-inflammatory macrophage activation. Epsin1 regulates DLL4 content in TEC-exosomes under high-glucose conditions. TEC-exosomes without Epsin1 inhibit N1ICD activation and iNOS expression in THP-1 cells. This indicates that Epsin1 may regulate communication between tubules and macrophages in diabetic nephropathy by facilitating the exosomal sorting of DLL4 and activating Notch1 [46].

Another study has shown that communication between tubular epithelial cells and macrophages creates a negative feedback loop via extracellular vesicles (EV), resulting in renal inflammation and apoptosis in diabetic nephropathy. To assess the effect of lipids on renal tubular epithelial cells (TECs), primary TECs from patients with diabetic nephropathy or HK-2 TECs were treated with lysophosphatidylcholine (LPC), leading to elevated EV release. Lipotoxicity-related TEC-derived extracellular vesicles (EVe) induce pro inflammatory phenotype in macrophages and stimulate the release of macrophage-derived extracellular vesicles (EVm). Additionally, EVm induced apoptosis in tubular epithelial cells injured by LPC. Leucine-rich α-2-glycoprotein 1 (LRG1)-enriched EVe activates macrophages through a TGFβR1-dependent mechanism, while tumor necrosis factor-related apoptosis-inducing ligand (TRAIL)-enriched EVm induces apoptosis in injured TECs via a death receptor 5 (DR5)-dependent process [47].

Various insults to the tubulointerstitial region, such as ischemia and proteinuria, activate the local renin–angiotensin system (RAS) in tubulointerstitial or infiltrating cells [48]. Angiotensin II (Ang II) has various effects, such as stimulating reactive oxygen species (ROS) and chemokines at inflammatory sites. Research indicates that Ang II can induce MCP-1 protein expression in mouse proximal tubular cells, a process inhibited by ROS, Ras, and NF-κB inhibitors. Both Angiotensin II type 1 and type 2 receptors (AT1R and AT2R) antagonists have been found to partially reduce MCP-1 expression. In mesangial cells, inhibitors of protein kinase and NF-κB were more effective in reducing acute Ang II-induced MCP-1 expression compared to ROS/Ras inhibitors. ROS-mediated signaling in proximal tubular cells may have a more significant role in Ang II-induced inflammatory responses than in mesangial cells [49]. On the other hand, the overactivity of the renin–angiotensin system in tissues is suppressed by angiotensin II type 1 receptor-associated protein (ATRAP), which inhibits receptor signaling. ATRAP is predominantly expressed in the kidney, with the highest levels found in tubules rather than glomeruli. Tubular ATRAP-mediated modulation of angiotensin II type 1 receptor signaling increases the accumulation of tubulointerstitial anti-inflammatory alternative activated macrophages, impacting glomerular manifestations of diabetic nephropathy through tubule-glomerular crosstalk [50].

### 3.3. Crosstalk between Macrophage and Mesangial Cells within Diabetic Kidney

In diabetic nephropathy, dysfunction of mesangial cells and activation of inflammatory pathways are prevalent. Research has shown that exosomes derived from macrophages treated with high glucose can trigger the activation of the NOD-, LRR-, and pyrin domain-containing protein 3 (NLRP3) inflammasome and induce autophagy deficiency in mesangial cells. These exosomes, when taken up by mesangial cells, lead to the activation of inflammatory cytokines and the NLRP3 inflammasome. Studies on C57BL/6 mice have demonstrated that injecting these high-glucose-stimulated macrophage-derived exosomes results in renal dysfunction and expansion of the mesangial matrix [51]. Furthermore, these exosomes activate glomerular mesangial cells through the TGF-β1/Smad3 pathway. Notably, exosomes from high-glucose-treated macrophages with TGF-β1 knockdown induced lower extracellular matrix and inflammatory factors in mesangial cells compared to the vector control, indicating the crucial role of TGF-β1 mRNA in exosomes in the interaction between macrophages and mesangial cells [52].

The primary glomerular abnormality in diabetic nephropathy is mesangial expansion, which reduces the filtration area and can lead to sclerosis and renal failure. Prior to the expansion and sclerosis of the mesangial extracellular matrix, there is a phase of phenotypic activation and temporary proliferation of glomerular mesangial cells, followed by significant infiltration of monocytes and macrophages into the glomerulus [53,54]. An in vitro study revealed that an increase in glucose concentration significantly enhanced hyaluronan synthesis and the formation of an extracellular hyaluronan matrix that promotes monocyte adhesion after mesangial cell division [55]. Monocytes were observed to primarily bind to hyaluronan-based structures in vitro, and abnormal deposits of hyaluronan were found in the glomeruli of diabetic rats [56].

TLR-4 and its endogenous ligand, myeloid-related protein 8 (MRP8), have been found to have increased mRNA levels in the glomeruli of diabetic mice, with MRP8 expression being enhanced by high glucose and palmitate treatment in cultured macrophages [57]. The elimination of MRP8 in myeloid cells has been shown to suppress the induction of macrophage-inducible C-type lectin (MINCLE) expression and reduce inflammatory changes in diabetic nephropathy. Intraglomerular crosstalk between mesangial cells and macrophages, mediated by MRP8-dependent MINCLE expression, may play a role in the inflammatory changes observed in glomerulonephritis [58,59].

### 3.4. Crosstalk between Macrophage and Immune Cells within Diabetic Kidney

Diabetic nephropathy, a complication associated with both Type 1 and Type 2 diabetes, is characterized by the presence of CD4+ and CD8+ T cells within renal tissue. These T cells engage with macrophages to regulate inflammation and renal damage [5,60]. Activated T cells can cause harm through direct cytotoxic mechanisms and indirectly by recruiting and activating macrophages. The proinflammatory cytokines released by T cells can trigger neighboring macrophages by inducing mesangial cell production of colony-stimulating factor-1 and MCP-1 [61]. Upon activation, macrophages release various substances such as nitric oxide, reactive oxygen species, IL-1, TNF-α, complement factors, and metalloproteinases, all of which contribute to renal injury [62,63]. T cells possess receptors for AGEs and can respond to them. Stimulation of CD4+ and CD8+ T cells by AGEs can result in the secretion of interferon-γ (IFN-γ), leading to heightened inflammation and oxidative stress in the diabetic kidney [64]. Furthermore, CD8+ cells may exhibit cytotoxic activity within the diabetic kidney. The cytokines and molecules produced in this context promote inflammation and further induce the expression of macrophage colony-stimulating factor and ICAM-1 in renal cells, thereby exacerbating renal injury [65,66]. B cells play a crucial role in humoral immunity and have functions that extend beyond antibody production. Within organs like the lung, liver, kidney, and urinary bladder, a distinct population of tissue-resident B cells, with a significant number being B-1a cells, can be found. These B cells are frequently located near macrophages, assisting in regulating their activation and behavior. B cells promote an anti-inflammatory macrophage phenotype by secreting interleukin-10. This interaction has implications for processes such as bacterial elimination in urinary tract infections and inducing immune tolerance in kidney transplant recipients [67,68]. The specific role of regulatory B cells in diabetic kidney disease, especially concerning macrophages, remain ambiguous.

Dendritic cells have traditionally been recognized as initiators of adaptive immune responses and share functional similarities with macrophages in maintaining renal homeostasis. The classification of renal mononuclear cell populations into distinct macrophages and dendritic cells is challenging due to the co-expression of common surface markers, such as CD11b, F4/80, and CD68 for macrophages, and CD11c, myosin heavy chain II, and CD80/86 for dendritic cells [69]. In cases of acute kidney injury, dendritic cells in the kidney initially play a protective role in limiting damage. However, in chronic kidney inflammation, dendritic cells undergo functional alterations that may contribute to the progression of kidney disease [70]. Resident mononuclear cells in the kidney, including macrophages and dendritic cells, possess components of the NLRP3 inflammasome and have the capacity to release mature pro-inflammatory cytokines. Consequently, renal mononuclear cells undergo caspase-1 dependent pyroptosis, a process that is implicated in exacerbating diabetic nephropathy [71,72].

## 4. Polarization of Macrophages within Diabetic Microenvironment

Macrophages display diversity in their characteristics and functions, which are influenced by the surrounding microenvironment. They play a dual role in both protecting the host and causing tissue damage, a delicate balance that is maintained. Macrophages can be categorized into two main subsets: the classical M1-type activation and the alternative M2-type activation. M1 macrophages, also known as the pro-inflammatory phenotype, are activated by various stimuli such as pathogen-associated molecular patterns (PAMPs) [73], danger-associated molecular patterns (DAMPs) [74,75], and pro-inflammatory cytokines like IL-1α [76], interferon-γ [77], and TNF-α [77,78]. These M1 polarized macrophages produce pro-inflammatory molecules, including IL-1, IL-6, IL-12, IL-23, iNOS, matrix metalloproteinase 12 (MMP12), and MINCLE [58,79,80,81]. On the other hand, M2 macrophages, which can originate from infiltrating monocytes or transition from M1 macrophages, have diverse functions that involve suppressing inflammation and promoting tissue repair [82,83,84,85]. M2 macrophages can be further classified into three subsets: M2a, M2b, and M2c, induced by different stimuli, such as IL-4, IL-13, immune complexes, TLR ligands, IL-1R ligands, IL-10, TGF-β, glucocorticoid, and other factors [86,87,88,89,90]. A novel subset of M2 macrophages, termed M2d or tumor-associated macrophages, is induced by specific co-stimulations [91,92]. The various subtypes of M2 macrophages exhibit both shared and unique functions. Specifically, M2a is involved in promoting wound healing and tissue fibrosis, serving as a tissue repair macrophage. On the other hand, M2b is associated with immunoregulation, while M2c is involved in immunosuppression, matrix remodeling, and tissue repair. M2b and M2c are commonly denoted as regulatory macrophages [93,94,95]. (To efficiently locate information in this section, readers are encouraged to refer to Figure 2.)

Despite the traditional classification of macrophages into M1 and M2 types, recent single-cell RNA sequencing analyses suggest that this classification is not always clear-cut, especially in dynamic processes like macrophage polarization in conditions such as diabetic kidney disease [8,9]. In diabetic environments, factors like high sugar levels hinder M2 polarization due to various conditions including ongoing pro-inflammatory interactions between macrophages and non-myeloid cells, compromised autophagy and mitophagy, reduced efferocytosis ability, lipotoxicity, and decreased SIRT6 expression. The primary factors that negatively impact M2 polarization in the diabetic microenvironment are as follows.

### 4.1. Impairments in the Autophagic Process, Particularly Mitophagy, within a High-Glucose Microenvironment

In hyperglycemic diabetic mice, elevated dynamin-related protein 1 (DRP-1) levels cause mitochondrial fragmentation. Meanwhile, levels of mitochondrial biogenesis-related proteins, such as Sirtuins (SIRT) 1, SIRT3, phosphorylated AMP-activated protein kinase (AMPK), and peroxisome proliferator-activated receptor gamma coactivator-1α (PGC-1α), decrease [96,97]. Diabetes decreases mRNA expression of mitochondrial DNA-encoded subunits in mitochondrial complexes I, III, IV, and V, as well as nuclear-encoded subunits in peritoneal macrophages. This reduction results in decreased ATP production and mitochondrial membrane potential, as well as an increase in mitochondrial ROS. M1-like macrophages have shorter punctate mitochondria compared to M2-like macrophages, with variations in levels of mitofusin-2 (MFN-2), DRP-1, ATP content, and mitochondrial ROS. Inhibition of mitochondrial ROS has been shown to hinder M1 phenotype polarization in RAW 264.7 cells under high-glucose conditions. Moreover, high-glucose environments reduce LC3II, BECN1, and ATG5 proteins and increase SQSTM1/P62 levels in macrophages, leading to impaired autophagosome clearance and lysosomal function. This disruption leads to elevated lysosomal pH and reduced levels of lysosomal-associated membrane protein 1 (LAMP1), endopeptidase cathepsin B (CTSB), and transcription factor EB (TFEB) in macrophages, particularly in M1-like macrophages, indicating impaired lysosomal function. The dysregulation of lysosome function and autophagic flux during macrophage polarization in diabetes results in increased iNOS-1 expression, leading to an M1 phenotype switch. These findings emphasize the impact of diabetes on macrophage polarization and mitochondrial function [98].

Studies on streptozotocin (STZ)-induced diabetic rats reveal an M1 phenotype and reduced mitophagy in renal macrophages. A positive correlation exists between iNOS and SQSTM1/P62 expression, while a negative correlation is observed with LC3 II. Electron microscopy reveals mitochondrial damage and decreased lysosomes in rats with diabetic nephropathy compared to control group. In vitro experiments with the RAW264.7 macrophage cell line show a shift to an M1 phenotype in high-glucose conditions, resulting in reduced mitophagy. Inhibition of mitophagy by 3-methyladenine (3-MA) promotes an M1 phenotype shift, while rapamycin, a mitophagy activator, suppresses the expression of M1 markers induced by high glucose and promotes M2 markers [99].

Mesenchymal stem cells (MSCs) have been demonstrated to prevent renal injuries in diabetic models by promoting the M2 phenotype in macrophages, improving mitochondrial biogenesis, and enhancing autophagy function [100,101]. These findings highlight the crucial role of autophagy and mitophagy in regulating the M1/M2 macrophage phenotype in diabetic nephropathy. They suggest that the inability to transition from the M1 to anti-inflammatory phenotype in diabetic macrophages is linked to deficiencies in PGC-1α-mediated mitochondrial biogenesis and impaired PGC-1α/TFEB-mediated lysosome-autophagy.

### 4.2. Impaired Efferocytosis and Expression of AIM2 Inflammasomes Impede the Resolution of Inflammation

Efferocytosis, the process by which macrophages clear apoptotic cells, is crucial for resolving inflammation by sequestering dying cells to prevent the release of intracellular DAMPs that can trigger inflammation [102,103]. This mechanism activates resolution signaling pathways through the breakdown of apoptotic cells within phagolysosomes. Nucleotides released from apoptotic cell DNA breakdown by DNase2a initiate a pathway involving DNA-dependent protein kinase catalytic subunit (DNA-PKcs), the mammalian target of rapamycin complex 2 (mTORC2)/rapamycin-insensitive companion of mTOR (RICTOR) complex. This leads to increased Myc expression and non-inflammatory macrophage proliferation via G protein-coupled receptor 132 (GPR132), known as efferocytosis-induced macrophage proliferation (EIMP). This process has been shown to boost the M2 macrophage population in different in vitro and mouse models, including zymosan-induced peritonitis [104], dexamethasone-induced thymocyte apoptosis [105], and atherosclerosis regression [106,107].

Researchers have used scRNAseq and ligand-receptor network analysis to reveal interactions between macrophages in diabetic kidney disease and other cells. Further analysis using protein–protein interaction analysis showed a significant link between the Rho GTPase ras-related C3 botulinum toxin substrate 1 (RAC-1) and macrophage efferocytosis. Overexpressing RAC1 enhances macrophage efferocytosis and reduces the inflammatory response in vitro. Conversely, reduced RAC1 expression was noted in macrophages treated with lipopolysaccharide in high-glucose conditions [108].

The RAGE plays a unique role in enhancing efferocytosis by interacting with phosphatidylserine (PS) on apoptotic cells. Reduced efferocytosis was observed in macrophages from RAGE-deficient mice and in normal macrophages treated with advanced glycation end products, which compete for RAGE binding. Overexpressing RAGE in human embryonic kidney 293 cells (HEK293) enhanced efferocytosis. In vivo studies showed reduced phagocytosis of apoptotic neutrophils by macrophages in RAGE-deficient mice [109]. Elevated AGEs in type 2 diabetes inhibit macrophage phagocytosis by disrupting RAC-1 activity and cytoskeletal rearrangement through RAGE/Rho kinase signaling in macrophages [110].

High-mobility group box 1 (HMGB1), a non-histone nuclear protein, has been identified as a molecule that disrupts efferocytosis. Upon cellular stimulation by TLR-4, HMGB1 undergoes extensive polyADP-ribosylation (PARylation) upon secretion. The translocation of HMGB1 from the nucleus to the extracellular space following TLR-4 activation can be decreased by inhibiting poly(ADP-ribose) polymerase-1 (PARP-1). Efferocytosis was impaired when macrophages or apoptotic cells were pre-incubated with HMGB1, which interacts with PS on apoptotic cells and RAGE on macrophages. The inhibitory effect on efferocytosis was more pronounced with PARylated HMGB1, which exhibited higher affinity for PS and RAGE compared to unmodified HMGB1. In a diabetic environment, both PARP1 and HMGB1 are increased [111,112,113,114], with PARylated HMGB1 showing a stronger inhibitory effect on macrophage activation during efferocytosis [115].

Regulatory T cells (Treg) enhance macrophage efferocytosis during inflammation resolution through transcellular signaling. In type 2 diabetes mellitus, Treg cells are more prone to apoptosis, decreasing their numbers and function. This can result in metabolic issues such as insulin resistance and a weakened anti-inflammatory role of Treg cells, potentially impeding efferocytosis in diabetes [116].

While efferocytosis is typically anti-inflammatory, macrophages can uptake leaked intracellular contents from necrotic cells, such as DNA, which can activate the absent-in-melanoma 2 (AIM2) inflammasome protein, leading to inflammation in chronic kidney disease. In a mouse model of unilateral ureteral obstruction (UUO), AIM2 deficiency reduced renal injury, fibrosis, and inflammation compared to wild-type (WT) littermates. Proinflammatory macrophages move along injured tubules, engulfing DNA from necrotic cells, and express active caspase-1, leading to the maturation of pro-IL-1β and pro-IL-18 [117]. DNA uptake occurs in vacuolar structures in recruited macrophages, not in resident CX3C motif chemokine receptor 1 (CX3CR1)+ renal phagocytes, contributing to chronic kidney injury [118]. AIM2 expression is present in glomeruli, tubules, and infiltrating leukocytes in diabetic or non-diabetic CKD patients’ kidneys, with AIM2 inflammasome activation mainly in macrophages [119]. Increased levels of AIM2 and circulating mitochondrial DNA, as well as various cytokines, were observed in type 2 diabetes [120,121,122]. Mitochondrial DNA from patients with type 2 diabetes mellitus can trigger AIM2 inflammasome-dependent caspase-1 activation, leading to the release of proinflammatory cytokines and promoting polarization to the M1 phenotype [121].

### 4.3. The Presence of Exosomes in DLL4 and the Disruption of microRNA Regulation Hinder the M2 Polarization Process

Extracellular vesicles (EVs) from human serum albumin (HSA)-stimulated tubular epithelial cells (HK-2 cells) can induce macrophage M1 polarization when co-cultured with macrophages in the presence of lipopolysaccharide (LPS). Through bioinformatic analysis, researchers found that microRNA (miR)-199a-5p is upregulated in EVs from HSA-induced HK-2 cells and in urinary EVs from diabetes patients with macroalbuminuria. Injection of EVs from HSA-induced HK-2 cells into diabetic mice via the tail vein led to M1 polarization of kidney macrophages, accelerating the progression of diabetic kidney disease (DKD) via miR-199a-5p action. The miR-199a-5p accelerates DKD progression by affecting the TLR-4 pathway through Klotho targeting, promoting macrophage M2 polarization [123]. Inhibiting glycolysis activation in macrophages with 2-Deoxy-d-glucose reduced the expression of inflammatory and fibrotic genes. Conversely, EVs from HSA-treated HK-2 cells were found to boost macrophage glycolysis. These EVs were found to increase macrophage glycolysis by stabilizing HIF-1α, leading to an inflammatory response [124]. Moreover, in extended high-glucose conditions, tubular epithelial cells release exosomes with DLL4, decreasing macrophage survival after M2 polarization. DLL4 triggers caspase 3/7-dependent apoptosis in M2 macrophages but not in M1 macrophages, with apoptosis controlled by DLL4 relying on NOTCH. Fully differentiated M2 macrophages become resistant to DLL4-induced apoptosis. Mechanistically, DLL4 selectively upregulates gene expression during M2 macrophage polarization, affecting NOTCH signaling, activity, and transcription. The pro-apoptotic effectors BCL2-associated X protein (BAX) and BCL2 antagonist/killer 1 (BAK1, also known as BAK), along with the BH3-only proteins BID and BIM, transmit the apoptotic signal induced by DLL4. Activation of the NOTCH signaling pathway by DLL4 inhibits M2 macrophage differentiation and promotes apoptosis [45].

MicroRNA (miRNA) plays a crucial role in maintaining cellular balance and regulating the expression of most protein-coding genes in the human genome. Several miRNAs have been linked to diabetes and kidney disease. A thorough literature review and bioinformatic analysis have consistently shown specific miRNA dysregulation in individuals with type 1 and type 2 diabetes mellitus [125,126]. Several microRNAs, like miR-9, miR-127, miR-155, and miR-125b, support M1 polarization, while others, such as miR-124, miR-233, miR-34a, let-7c, miR-132, miR-146a, and miR-125a-5p, induce M2 polarization [127,128]. Studies have shown that miRNA-21 is elevated in diabetic nephropathy, playing a role in the disease by interacting with specific genes and signaling pathways. This interaction triggers various biological processes, including epithelial-to-mesenchymal transition, extracellular matrix deposition, cytoskeletal remodeling, inflammation, podocyte pyroptosis, and fibrosis [129,130]. Despite its role in promoting M1 polarization in chronic obstructive pulmonary disease and diabetic wounds [131,132], the effect of miR-21 on polarization in diabetic kidney disease is unclear. MiR-146a is known to protect against kidney injury and act as an anti-inflammatory agent in diabetic nephropathy by promoting M2 polarization [133,134,135]. However, decreased levels of miR-146a in diabetes can impede M2 polarization in diabetic kidney disease, leading to the loss of its anti-inflammatory and antifibrotic effects [136].

### 4.4. Elevated Blood Sugar Levels Lead to the Suppression of SIRT6, Thereby Impeding the Process of M2 Macrophage Polarization

The sirtuin family proteins, consisting of SIRT1-7, are a group of β-NAD+- or NAD+-dependent signaling proteins known for their involvement in metabolic regulation, including processes such as inflammation, metabolism, oxidative stress, and apoptosis [137]. Several research studies have highlighted the impact of sirtuins on metabolic pathways and the differentiation of macrophages, influencing inflammation in various contexts [138,139,140,141,142,143,144,145,146,147]. Specifically, SIRT6 has been shown to offer renal protection in diabetes by modulating macrophage polarization. In a diabetic nephropathy rat model, the upregulation of SIRT6 prompted macrophages to transition into the M2 phenotype, thereby protecting podocytes from damage caused by elevated glucose levels. However, under high-glucose conditions, podocyte apoptosis increased when podocytes overexpressing SIRT6 were co-cultured with macrophages. The protective mechanisms of SIRT6 involve the suppression of apoptosis, upregulation of Bcl-2 and CD206 expression, and downregulation of Bax and CD86 expression. By promoting M2 macrophages, SIRT6 shields podocytes from injury in a simulated diabetic kidney microenvironment [148]. Nevertheless, high glucose levels induce the transformation of macrophages into the M1 phenotype and trigger podocyte apoptosis in a dose-dependent manner, while also reducing SIRT6 expression and hindering macrophage M2 polarization [149,150,151,152].

## 5. The Participation of Macrophages in the Development of Renal Fibrosis in Diabetic Kidney Disease

Numerous research studies have highlighted the significant involvement of M2 macrophages in promoting kidney fibrosis, despite their well-known anti-inflammatory properties and contribution to resolution processes [7,153,154,155]. Tubulointerstitial injury, a common characteristic of renal disorders, is exacerbated by interstitial fibrosis, which plays a pivotal role in the progression and deterioration of kidney function. Fibroblasts, essential for collagen production, are instrumental in extracellular matrix (ECM) accumulation. Various subtypes of fibroblasts have been distinguished based on their sources, including resident fibroblasts, bone marrow-derived fibroblasts, epithelial cells, endothelial cells, and pericytes.

### 5.1. Epithelial to Mesenchymal Transition vs. Macrophage–Myofibroblast Transition

At the start of injury, M1 macrophages are the main cell type involved. Macrophages release pro-inflammatory cytokines that worsen injuries, intensify the inflammatory response, and promote the proliferation of myofibroblasts and the recruitment of fibrocytes [156]. These M1 macrophages activate metalloproteinases that break down the ECM, facilitating processes like epithelial-to-mesenchymal transition (EMT) or endothelial-to-mesenchymal transition (EndoMT) [157]. Additionally, supernatants from foam cells derived from M1 macrophages can induce endothelial-to-mesenchymal transition (EndoMT) by increasing the expression of C-C motif chemokine ligand 4 (CCL-4) [158]. On the other hand, M2 macrophages release factors like TGF-β1, fibroblast growth factor 2 (FGF-2) [159], platelet-derived growth factor (PDGF) [160], or galectin-3 [161] to promote the proliferation, viability, and activation of myofibroblasts, leading to increased ECM accumulation [162]. M2 macrophages can activate myofibroblasts through various sources such as EMT, EndoMT, pericytes, or mesangial cells by releasing IL-1, matrix metalloproteinase-9 (MMP-9), and TGF-β [162,163,164].

In addition to indirectly promoting fibrosis by the recruitment, proliferation, and activation of fibroblasts, macrophages can also contribute to fibrosis directly by transforming into myofibroblasts through a process called macrophage–myofibroblast transition (MMT). Co-expression of macrophage and myofibroblast antigens identifies the MMT process in human and experimental fibrotic kidney disease [165]. Biopsies from 58 patients with various kidney diseases were examined to identify myofibroblasts originating from bone marrow-derived monocytes/macrophages. The analysis showed a significant presence of cells expressing both macrophage (CD68) and myofibroblast (α-smooth muscle actin, α-SMA) markers. MMT cells were found in active fibrotic lesions but were mostly absent in acute inflammatory or sclerotic lesions, suggesting their involvement in progressive renal fibrosis. MMT cells showed a predominant M2 phenotype in human and mouse renal fibrosis. Additionally, selectively depleting myeloid cells in mice with diphtheria toxin significantly reduced macrophage infiltration and MMT cells in the obstructed kidney. The reduction in macrophages led to a decrease in myofibroblasts and collagen deposition, highlighting the role of macrophages in tissue fibrosis [166]. The transition to MMT was primarily seen in M2-type macrophages and was controlled by TGF-β/Smad3 signaling. The absence of Smad3 in the bone marrow of GFP+ chimeric mice hindered the transition of M2 macrophages into MMT cells, impacting the development of renal fibrosis. In vitro experiments using Smad3-deficient bone marrow macrophages demonstrated that Smad3 is crucial for TGF-β1-induced MMT and collagen synthesis. The TGF-β/Smad3 signaling pathway regulates the conversion of bone-marrow-derived macrophages into myofibroblasts in tissue fibrosis [167].

MMT cells were identified as a significant source of collagen-producing fibroblasts in fibrotic kidneys, making up over 60% of α-SMA+ myofibroblasts in chimeric mice following the UUO procedure. A detailed study on genetically modified mice revealed that half of the myofibroblast population originates from local resident fibroblasts through proliferation. The remaining non-proliferating myofibroblasts differentiate from bone marrow (35%), EndoMT program (10%), and EMT program (5%). These studies emphasize the significance of MMT in renal fibrosis [163,167].

### 5.2. M2 Macrophage, to Be or Not to Be? That Is a Question

The circumstances in which M2 macrophages contribute to anti-inflammatory responses and tissue repair, or promote renal fibrosis, remain unclear. M2 macrophages are categorized into subtypes such as M2a, M2b, M2c, and M2d, each with distinct markers and triggers [168]. In vitro research indicates that M2a and M2c macrophages provide protection by inhibiting CD4+ T-cell proliferation and deactivating host macrophages. In vivo studies have shown that M2a and M2c macrophages can protect against renal structural and functional damage. M2c macrophages, possibly because of their B7-H4-dependent capacity to convert naive T cells into regulatory T cells (Tregs), are more efficient than M2a macrophages in decreasing Adriamycin-induced renal damage. The superior protective effects of M2c compared to M2a are backed by regulatory T cell depletion [93]. In a mouse model of unilateral ureteral obstruction (UUO), M2a macrophages contribute to disease progression through MMT cells in renal fibrosis, while M2c macrophages demonstrate potent anti-inflammatory functions, promote tissue repair, and are inhibited. The level of TGF-β signaling is crucial for determining the distinct polarization of M2a and M2c. Excessive TGF-β stimulation induces M2a macrophages to undergo MMT, while moderate TGF-β stimulation promotes the polarization of M2c macrophages. By regulating the activating transcription factor 6 (ATF6)/TGF-β/Smad3 signaling axis in macrophages and adjusting TGF-β levels, it is possible to guide their polarization towards the M2c phenotype and inhibit excessive MMT polarization [169]. However, another study suggests that polarized M2c macrophages promote the epithelial-to-mesenchymal transition of human renal tubular epithelial cells [170]. The functions of M2b macrophages in different diseases encompass both protective and harmful effects. The precise involvement of M2b macrophages in kidney disease remains ambiguous, yet they could potentially serve as a crucial factor in the onset and advancement of autoimmune conditions [95]. M2d macrophages, also referred to as tumor-associated macrophages (TAM), are characterized by elevated IL10 expression and reduced IL12 levels. TAMs impede the immune response of anti-tumor T cells, consequently fostering a state of tumor tolerance [171].

A research study analyzed alterations in transcription factors during M2 macrophage polarization and their target genes to construct a gene co-expression network centered on transcription factors. The process of M2 polarization was classified into five distinct cell states. The study revealed that the shift of M2 macrophages from state 1 (CD206-CD68- M2 macrophages) to state 5 (CD206+CD68+ M2 macrophages) is the primary pro-fibrotic mechanism. The significant expression of fibrosis-promoting genes in CD206+CD68+ M2 macrophages indicates their potential to transform into fibrocytes. Further research is required to determine if this classification of M2 macrophages can effectively target their pro-fibrotic activity [172]. The presence of anti-inflammatory properties and the potential for fibrosis formation create ambiguity in using M2 macrophages for treating kidney disorders.

## 6. Proposals for the Treatment of Diabetic Nephropathy through the Modulation of Macrophages

Macrophages are significantly involved in the pathogenesis of diabetic nephropathy, prompting investigations into their potential therapeutic applications for this condition. One proposed intervention involves mitigating macrophage activation and involvement in the inflammatory cascade. Alternatively, another strategy focuses on leveraging macrophages’ capacity to regulate inflammatory responses and facilitate renal tissue repair. Subsequent sections will delve into these therapeutic considerations. (To efficiently locate information in this section, readers are encouraged to refer to Figure 3.)

### 6.1. Decrease Recruitment and Activation of Monocytes

The primary strategy for reducing macrophage activity in diabetic kidney disease involves limiting monocyte recruitment from the bloodstream and inhibiting macrophage activation by disrupting cell interactions or reducing activators. MCP-1, also known as CCL2, is a critical member of the CC chemokines family that plays a key role in the inflammatory response by attracting or enhancing the expression of other inflammatory factors or cells. Increased local production of MCP-1 is associated with glomerular injury in diabetic nephropathy through the recruitment and activation of macrophages. The presence of AGEs in a high-glucose environment can stimulate the production of chemokines in mesangial cells and podocytes [41]. Various compounds, such as mNOX-E36-3′PEG, loganin, TAK1 inhibitor, or CCR2 blocker, have been identified to reduce macrophage infiltration and activation by inhibiting the MCP-1/CCR2 axis in diabetic nephropathy. This leads to decreased expression of extracellular matrix components like fibronectin, type IV collagen, and TGF-β, normalization of IL-12 and IL-10 production, and a reduction in the number of infiltrating macrophages in renal tissue [173,174,175,176].

Several medications prescribed to individuals with diabetes to address metabolic issues have been found to decrease MCP-1 expression. The renin–angiotensin system plays a significant role in regulating MCP-1 expression locally, either directly or by affecting glomerular hemodynamics. Medications such as the angiotensin-converting enzyme inhibitor enalapril or the AT1 receptor antagonist candesartan have been effective in inhibiting the time-dependent increase in MCP-1 expression, which is closely associated with reductions in proteinuria and the number of glomerular macrophages in diabetic rats [24]. Pioglitazone, a medication used to manage blood sugar levels, has been shown to reduce macrophage presence and NF-κB activation in renal tissues in studies with diabetic rats [177]. Additionally, medications like pravastatin, fenofibrate, and aldosterone blockade have demonstrated potential in inhibiting macrophage recruitment and associated cytokines in diabetic mice, thereby preventing diabetic nephropathy progression [178,179,180]. Another approach to reduce involvement is by downregulating adhesion molecules, such as ICAM-1, which has shown promising results in decreasing albuminuria and leukocyte presence in diabetic mice [6]. Anthocyanin-rich purple corn extract (PCA) has been found to suppress monocyte activation and macrophage infiltration by inhibiting specific pathways [181]. Furthermore, the administration of 25(OH)D3 has been linked to decreased monocyte ER stress and SR-A1 expression, suggesting a potential role in regulating adhesion and movement in individuals with diabetes [182].

Combining specific anti-inflammatory agents like MIF inhibitor, Bronton’s tyrosine kinase inhibitor, cholecystokinin octapeptide (CCK-8) agonist, and β-arrestin2 agonist have shown promise in suppressing inflammatory signaling pathways and inhibiting inflammatory macrophage activation, offering potential implications for managing diabetic kidney disease [183,184,185,186]. Targeting MIF, a key factor in inflammation frequently elevated in diabetic kidney conditions, has shown promise in mitigating various aspects of diabetic nephropathy, making it a potential therapeutic strategy [183,187].

### 6.2. Adoption of Macrophage Polarization toward M2 Phenotype

The M2 macrophage is known for its anti-inflammatory properties and its involvement in tissue repair. Increasing M2 macrophage polarization may improve diabetic kidney disease.

In diabetic rats with nephropathy, there was an increase in infiltrating macrophages showing the M1 phenotype. Following treatment with calcitriol, there was a reduction in the heightened levels of iNOS and TNF-α expression. Prolonged administration of calcitriol can activate M2 macrophages. The CD163 to CD68 ratio, representing the M2 macrophage proportion, increased approximately 2.9 times after calcitriol treatment. In vitro experiments also showed that 1,25-dihydroxyvitamin D3 facilitated the conversion of M1 macrophages induced by high glucose into an M2 phenotype. Vitamin D plays a role in reducing macrophage infiltration, suppressing M1 macrophage activation, and enhancing the M2 macrophage phenotype to protect against podocyte injury [188]. The active form of vitamin D regulates macrophage M1/M2 phenotypes and prevents the transition of macrophages to the M1 phenotype via the STAT-1/triggering receptor expressed on myeloid cells 1 (TREM-1) pathway [189].

Recent studies have demonstrated the impact of sodium–glucose transport protein 2 (SGLT2) inhibitors on macrophage adaptability. A research study discovered that prolonged administration of dapagliflozin resulted in significant improvements in proteinuria, histomorphological damage, oxidative stress, and macrophage infiltration in the kidneys of diabetic mice induced by streptozotocin. This treatment was found to reduce renal inflammation and fibrosis by inhibiting the HMGB1/TLR-2/4/NF-κB signaling pathway. Furthermore, dapagliflozin reduced HMGB1 expression and downstream TLR-2/4/NF-κB signaling proteins in human proximal tubular (HK-2) cells exposed to high glucose and lipids, as well as in HMGB1-stimulated RAW264.7 cells following IL-1β stimulation. These findings confirm the anti-inflammatory properties of dapagliflozin and its ability to modulate the balance of M1/M2 macrophages [190]. In specific inflammatory conditions such as diet-induced obesity and LPS-stimulated macrophage models, empagliflozin has been observed to decrease the presence of pro-inflammatory M1 macrophages by inhibiting IKK/NF-κB, MKK7/JNK, and JAK2/STAT1 pathways, while simultaneously promoting the presence of anti-inflammatory M2 macrophages. This mechanism ultimately results in a reduction in tissue inflammation [191,192]. Furthermore, empagliflozin has shown significant inhibition of genes associated with fibrosis promotion and macrophage polarization, thereby impeding the formation of profibrotic CD206+CD68+ M2 macrophages through the regulation of mitophagy and mTOR pathways [172].

Other compounds like hyperoside [193], astragalus mongholicus Bunge and Panax notoginseng formula (A&P) [194], thalidomide [195], trichosanthes kirilowii lectin (TKL) [196], and schisandrin C [197] have been found to decrease renal inflammation in mice with type 2 diabetes by modulating macrophage polarization, facilitating the transition from M1 to M2 polarization.

Mesenchymal stem cells (MSCs) have demonstrated the ability to prevent renal injuries in diabetic models by regulating the immune system. MSCs induced the M2 macrophage phenotype and protected diabetic mice from renal injuries through a macrophage-dependent mechanism. The activation of TFEB by MSCs promotes M2 polarization, restoring lysosomal function and autophagy activity in macrophages. Moreover, transferring macrophages from diabetic mice treated with MSCs or macrophages cocultured with MSCs in vitro to DN mice improved renal function. The protective role significantly decreased when TFEB was suppressed in macrophages [100]. Additionally, mesenchymal stem cells (MSCs) communicate with their microenvironment through mitochondrial exchanges. Through a coculture system of MSCs and macrophages, the transfer of MSC-derived mitochondria into macrophages enhanced mitochondrial functions and promoted M2 polarization. This transfer of mitochondria from MSCs activated PGC-1α-mediated mitochondrial biogenesis. In high-glucose-induced macrophages, the interaction between PGC-1α and TFEB enhances lysosome–autophagy, crucial for eliminating damaged mitochondria. The enhancement of mitochondrial bioenergy in macrophages boosts their capacity to combat inflammatory responses. The immune-regulatory activity of mitochondria from MSCs was notably reduced in macrophages with PGC-1α knockdown. In diabetic nephropathy mice, transferring mitochondria from MSCs, adopted by macrophages, reduced inflammation and improved kidney function. The kidney-protective effects of mitochondria-transferred macrophages were lost when the MSC-derived mitochondria were impaired with rotenone. Mitochondria-transferred macrophages exhibited similar results after transfection with si-*pgc-1α* before administration [101]. Additionally. Umbilical cord (UC)-MSCs can shift macrophage polarization from pro-inflammatory M1 to anti-inflammatory M2 phenotype. Mechanistically, miR-146a-5p was significantly downregulated and negatively correlated with renal injury in diabetic nephropathy (DN) rats, as shown by high-throughput RNA sequencing. UC-MSCs-derived miR-146a-5p promotes M2 macrophage polarization by inhibiting the TNF receptor-associated factor 6 (TRAF6)/STAT1 signaling pathway [135].

The control of gene expression through non-coding RNAs, such as microRNAs (miRNAs), plays a crucial role in numerous cellular functions, and dysregulation of these primary gene expression regulators can lead to the onset of diseases. For instance, the activation of NF-κB has the ability to induce macrophage polarization into either M1 or M2 phenotypes during the progression of various diseases. miR-223 has been identified as a potential modulator of the spatial and temporal activation of NF-κB, thereby serving as a significant regulator of macrophage plasticity. Targeting mRNA may offer a strategy to mitigate these pathophysiological processes [198,199,200].

### 6.3. Utilizing Ex Vivo Macrophage Transfusion for the Treatment of Diabetic Kidney Disease

The ex vivo administration of M2 macrophages can enhance their anti-inflammatory properties, showing promise as a therapeutic strategy for chronic inflammatory renal conditions. In a study using a diabetic mouse model, macrophages were isolated from splenocytes and polarized with IL-4 and IL-13 to induce a protective phenotype. The M2 macrophages were transfused into mice before diabetes onset. The M2 macrophages transferred were observed to accumulate in the kidneys for up to 10 weeks after streptozotocin (STZ) treatment. Compared to diabetic mice in the control group, diabetic mice treated with M2 macrophages exhibited decreased tubular atrophy, glomerular hypertrophy, and interstitial expansion in their kidneys. M2 macrophages were found to protect against interstitial fibrosis and suppress its development [201]. This protective effect was also seen in severe combined immunodeficiency (SCID) mice with Adriamycin nephropathy, a model of chronic inflammatory renal disease similar to human focal segmental glomerulosclerosis. Macrophages from BALB/c mice spleens were stimulated to induce M1 or M2 macrophages. M2 macrophages showed a protective effect against histological and functional kidney damage [202].

However, the introduction of bone-marrow-derived M2 macrophages did not improve renal function or reduce renal damage in Adriamycin nephropathy, unlike splenic M2 macrophages. In vitro studies showed that bone marrow and splenic M2 macrophages had comparable regulatory and suppressive functions. In the inflamed kidney, bone marrow M2 macrophages exhibited decreased suppressive characteristics compared to splenic M2 macrophages, which was linked to their increased proliferation. The proliferation of bone marrow M2 cells in vivo was linked to the expression of macrophage colony-stimulating factor (M-CSF) by tubular cells in the inflamed kidney. Inhibiting M-CSF restricted the proliferation of bone marrow M2 cells and prevented their phenotypic transition, leading to an inflammatory rather than a regulatory phenotype and function. Consequently, bone marrow M2 macrophages did not protect against structural and functional renal damage in murine Adriamycin nephropathy, in contrast to splenic M2 macrophages [203].

A similar challenge of macrophage plasticity was also encountered when trying to use M2 macrophages to treat the UUO model. By genetically modifying macrophages through transfection with an adenoviral vector containing neutrophil gelatinase-associated lipocalin (NGAL) to induce NGAL overexpression, the resulting macrophages showed a stabilized phenotype with reduced susceptibility to phenotypic switching. The introduction of macrophages with elevated NGAL levels was linked to reduced kidney interstitial fibrosis and inflammation, mainly because the transplanted macrophages retained their phenotype and function [204]. Genetically modified macrophages stabilized by NGAL maintained their M2 phenotype and showed therapeutic advantages in decreasing albuminuria and renal fibrosis in a diabetic mouse model. This treatment resulted in an increase in the anti-inflammatory cytokine IL-10, a decrease in certain pro-inflammatory cytokines, a reduction in M1 glomerular macrophages, podocyte loss, and decreased renal TGF-β1 levels [205]. Macrophages’ adaptability poses a challenge to effective macrophage therapy for chronic nephropathies, which could be overcome by introducing *lipocalin-2*.

### 6.4. Strategies to Inhibit the Transition of Macrophages into Myofibroblasts

Several signaling pathways, such as the TGF-β/Smad3/Src pathway, are involved in MMT [167,206,207,208,209,210]. In the fibrotic kidney, recruited Smad3−/− macrophages failed to differentiate into myofibroblasts. The study showed a notable decrease in collagen I deposition and α-SMA, indicating a reduction in renal fibrosis in the UUO kidney [167]. Smad3−/− chimeric mice show an antifibrotic effect, indicating the involvement of bone marrow-derived macrophages in renal fibrosis development through MMT. Deletion of the Smad7 gene, a negative inhibitor, results in heightened TGF-beta/Smad2/3-dependent renal fibrosis in the UUO model [211]. Both AGE and Ang II can activate Smad signaling through TGF-β dependent and independent mechanisms in diabetes [212,213]. Asiatic acid acts (AA) as a Smad7 agonist, inhibiting TGF-β/Smad3-mediated renal fibrosis by enhancing Smad7 expression. Naringenin (NG) inhibits Smad3, preventing renal fibrosis by blocking Smad3 phosphorylation and transcription. The combination of AA and NG had an additive effect on inhibiting renal fibrosis by blocking Smad3 and upregulating Smad7 [214]. Disruption of Smad3 impacts immune function, resulting in compromised mucosal immunity and decreased T cell response to TGF-β. Targeting Pou4f1, a neural transcription factor and Smad3 target gene, may be a potential therapy for chronic kidney disease with progressive renal fibrosis [215]. Another research investigation employed specialized nanotechnology to simultaneously deliver an endoplasmic reticulum stress (ERS) inhibitor (Ceapin 7) and dexamethasone, successfully regulating the ATF6/TGF-β/Smad3 signaling pathway in macrophages. This approach controlled the extent of TGF-β activation in macrophages, facilitating their differentiation towards the M2c phenotype while inhibiting excessive MMT polarization [169].

JAK3/STAT6 signaling, Jumonji domain containing 3 (Jmjd3)/interferon regulatory factor 4 (IRF4) axis, and natural killer T cell/IL-4 signaling can activate myeloid fibroblasts, promoting the transition of M2 macrophages into myofibroblasts in kidney fibrosis in obstructed mouse kidneys [216,217,218]. Myeloid Jmjd3 deletion or Jmjd3 inhibitor reduced the expression of IRF4, α-smooth muscle actin, and fibronectin, inhibiting MMT in cultured monocytes exposed to IL-4. These pathways may contribute to diabetic kidney disease. Manipulating these pathways may impede MMT and related renal fibrosis.

### 6.5. Timing and the Corresponding Microenvironment Play Important Roles

The utilization of cell therapy involving macrophages, specifically M2 macrophages, shows promise in the management of chronic kidney disease. However, challenges arise from findings in research on diabetic wound healing, pressure ulcer healing, and post-myocardial infarction remodeling. Notably, a study demonstrated that local injection of activated macrophage suspension, rather than M2 macrophages, in elderly patients with stage III and IV pressure ulcers resulted in a significantly higher percentage of fully closed wounds [219]. Conflicting outcomes were observed in the early stages of post-myocardial infarction treatment with M1 macrophages instead of M2, despite variations in administration route and dosage [220,221]. Unregulated activation of macrophages can impede wound healing, leading to chronic wounds [222]. Conversely, early administration of autologous M1 macrophages on the first day of injury can enhance wound healing and normalize diabetes-induced skin repair issues. However, the administration of ex-vivo-activated murine macrophages, whether M2a or M2c, during the initial inflammatory phase following injury, resulted in impaired skin healing and delayed re-epithelialization in a diabetic mouse model [223,224]. One speculated hypothesis is that factors in the microenvironment may influence these effects. For instance, a decrease in MMPs levels could potentially contribute to the delayed healing phenotype in wounds treated with M2a and M2c [225]. Studies on the injury and regeneration of sensory hair cells in zebrafish have revealed an anti-inflammatory activation sequence in macrophages, highlighting the synergistic involvement of IL-10 signaling and IL-4 signaling macrophages in regeneration. This sequence involves macrophages initially displaying a glucocorticoid activation signature, followed by IL-10 signaling, and ultimately the induction of oxidative phosphorylation through IL-4/polyamine signaling in zebrafish neuromasts [226]. A similar sequence has been identified in human wound healing through transcriptome analysis, where IL-10-stimulated (M2c) macrophage-related genes were upregulated early after injury, akin to M1-related genes, while IL-4-stimulated (M2a)-related genes were expressed at later stages or downregulated post-injury [227]. This temporal sequence plays a crucial role in healing and repair, suggesting a potential synergistic relationship between these macrophage subtypes that warrants further investigation.

## 7. Conclusions

In diabetic nephropathy, macrophages play a crucial role in perpetuating inflammation through interactions with lymphocytes and other renal cells via extracellular vesicles, miRNA, and immune mediators. This inflammatory cascade is sustained by a feedback loop. While macrophages possess the capacity to transition into anti-inflammatory M2 macrophages for tissue repair, chronic exposure to high glucose levels disrupts this process. Consequently, macrophages lose their ability to sequentially modulate inflammation, anti-inflammation, and repair functions, leading to a dysregulated state characterized by coexistence and competition among macrophage subtypes. This dysregulation can contribute to organ degeneration and fibrosis. Given the central role of macrophages in chronic inflammation, a comprehensive understanding of their diverse functions and effective regulation may offer promising avenues for advancing therapeutic strategies for diabetes and chronic kidney disease. However, it is important to consider the potential systemic consequences of modulating macrophage function, as these cells play crucial roles in immune regulation and tissue homeostasis throughout the body. As previously stated in the article, the disturbance of Smad3 has been shown to affect immune function, leading to impaired mucosal immunity and reduced T cell reaction to TGF-β. The primary adverse effects associated with TGF-β inhibitors include increased liver enzyme levels, proteinuria, and anemia [215,228]. While there is promising potential for macrophage-based cell therapy in the treatment of diabetic kidney disease based on current understanding of macrophage involvement in the condition, only one clinical trial utilizing macrophage cell therapy for acute kidney injury has been identified on ClinicalTrials.gov thus far [229]. With the ongoing progress in scientific research and precision medicine, there is a growing emphasis on integrating multi-omics and high-throughput technologies to discover new molecular targets, create more precise macrophage modulators, and plan suitable clinical trials. These advancements are expanding the potential applications of macrophages in emerging therapeutic fields. Prior to the arrival of the anticipated future, engaging in physical activity may be a prudent and beneficial option to enhance oxygen levels, promoting the polarization of macrophages in support of disease regression [230,231,232].

## Figures and Tables

**Figure 1 ijms-25-04350-f001:**
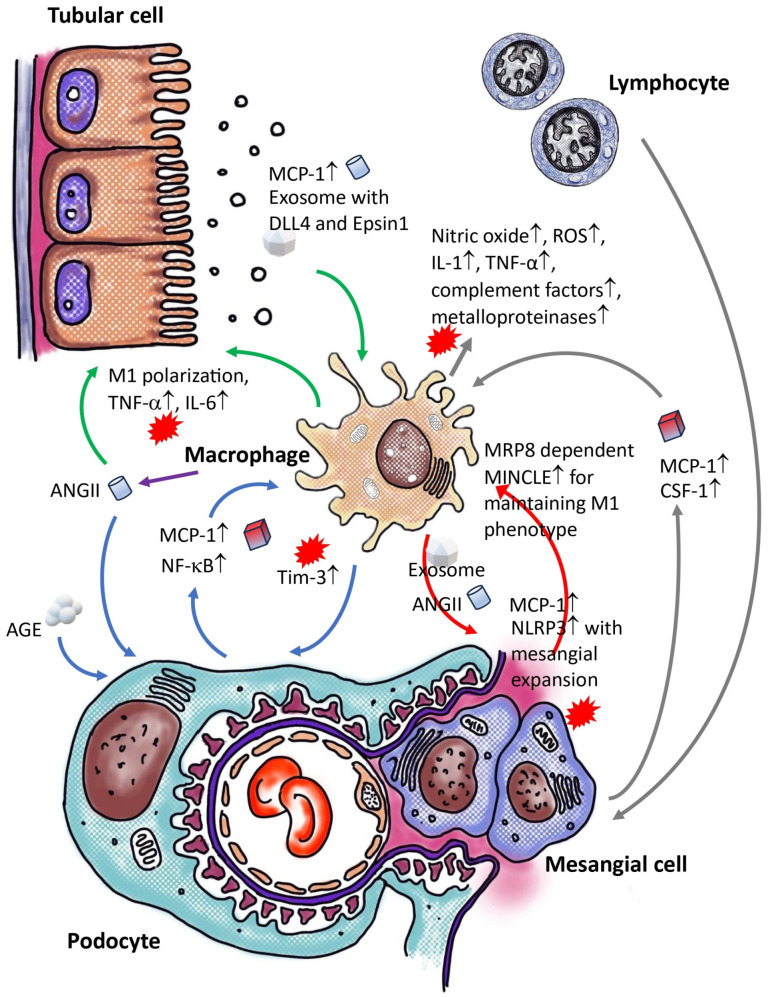
Crosstalk between macrophages and intrarenal non-myeloid cells. Macrophages engage in communication with various non-myeloid cells within the kidney, creating an amplifying loop that contributes to the advancement of kidney disease. The distinct colors of the lines in the illustration represent different instances of this crosstalk. AGE: advanced glycation end-products; ANGII: angiotensin II; CSF-1: macrophage colony-stimulating factor; DLL4: delta-like ligand 4; IL-1: interleukin-1; IL-6: interleukin-6; MCP-1: monocyte chemoattractant protein-1; MINCLE: macrophage-inducible C-type lectin; MRP8: myeloid-related protein 8; NF-κB: nuclear factor-kappa B; NLRP3: pyrin domain-containing protein 3; ROS: reactive oxygen species; Tim-3: T cell immunoglobulin and mucin domain 3; TNF-α: tumor necrosis factor α; Symbol ↑ equal to “increased”; Symbol ↓ equal to “decreased”.

**Figure 2 ijms-25-04350-f002:**
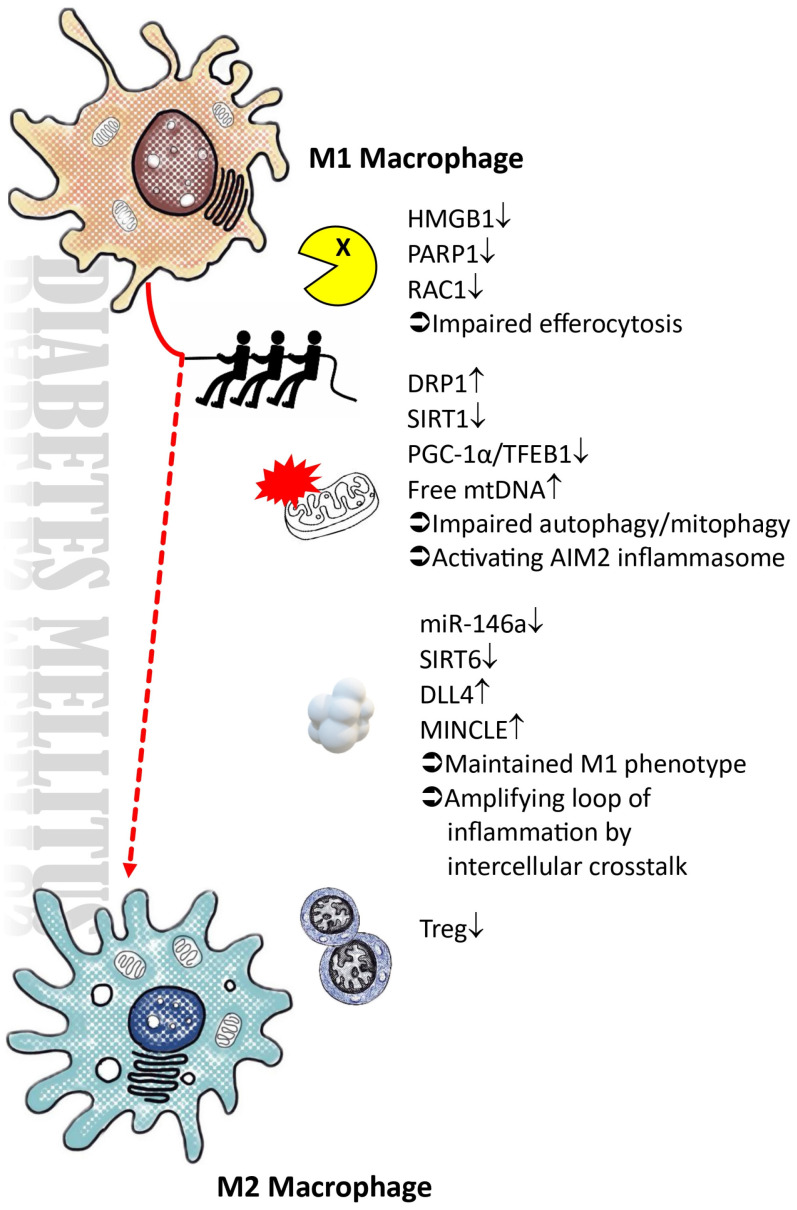
Adverse conditions that hinder the polarization of M2 macrophages within the diabetic microenvironment. DLL4: delta-like ligand 4; DRP-1: dynamin-related protein 1; HMGB1: high-mobility group box 1; MINCLE: macrophage-inducible C-type lectin; PARP1: Poly [ADP-ribose] polymerase 1; PGC-1α: peroxisome proliferator-activated receptor-gamma coactivator; RAC1: ras-related C3 botulinum toxin substrate 1; SIRT: sirtuin; Treg: regulatory T cell; Symbol ↑ equal to “increased”; Symbol ↓ equal to “decreased”.

**Figure 3 ijms-25-04350-f003:**
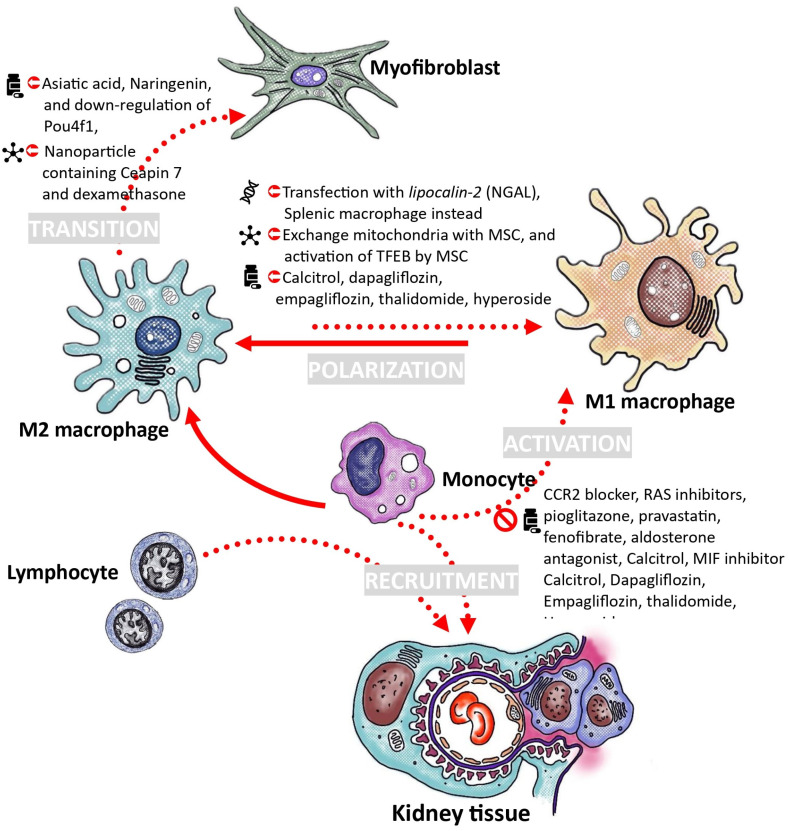
Utilizing macrophages as a therapeutic approach for the management of diabetic kidney disease. The macrophage-based approaches for managing diabetic kidney disease primarily focus on reducing monocyte recruitment and macrophage activation, facilitating the shift towards the anti-inflammatory M2 subtype, utilizing ex vivo macrophage cell therapy, and inhibiting the transition of macrophages into myofibroblasts. Solid line represents the direction of development, while the dashed line represents the direction to avoid. CCR2: C-C chemokine receptor type 2; TFEB: transcription factor EB; MIF: macrophage migration inhibitory factor; MSC: mesenchymal stem cell; NGAL: neutrophil gelatinase-associated lipocalin; RAS: renin–angiotensin system.

## Data Availability

Not applicable.

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
