# Peer review of "Targeting Macrophages: Therapeutic Approaches in Diabetic Kidney Disease"

_ijms, 2024, doi:10.3390/ijms25084350_

Round 1

Reviewer 1 Report

Comments and Suggestions for Authors

The authors discussed the role of macrophages in the pathogenesis and potential treatment of diabetic kidney disease. In the early stages of diabetic nephropathy, macrophages are recruited to the kidneys by chemokines and adhesion molecules, where they become activated by the hyperglycemic environment. These activated macrophages then interact with various renal cells, such as podocytes, epithelial cells, mesangial cells, and other immune cells, through extracellular vesicles, miRNA, and immune mediators, creating an inflammatory cascade that perpetuates kidney damage. The article also highlights the impairment of macrophage polarization from the pro-inflammatory M1 phenotype to the anti-inflammatory and reparative M2 phenotype in the diabetic milieu, which is attributed to factors such as compromised autophagy, reduced efferocytosis, and suppressed SIRT6 expression.

Comments:

  1. Although the article mentions the existence of different macrophage subsets (M1, M2a, M2b, M2c, and M2d), it does not provide an in-depth discussion on their specific roles and the potential impact of their heterogeneity on the pathogenesis and treatment of diabetic nephropathy.
  2. The article could benefit from a more detailed discussion on the temporal changes in macrophage phenotypes and functions throughout the progression of diabetic kidney disease, as this information may have important implications for the timing of therapeutic interventions.
  3. While the article suggests several macrophage-targeted therapeutic approaches, it does not provide a comprehensive assessment of their translational potential, such as discussing the feasibility, safety, and efficacy of these strategies in clinical settings.
  4. The article primarily focuses on the local effects of macrophages within the kidney. However, it is important to consider the potential systemic consequences of modulating macrophage function, as these cells play crucial roles in immune regulation and tissue homeostasis throughout the body.
  5. Although the article mentions the interaction between macrophages and T cells, it does not provide a detailed discussion on the complex interplay between macrophages and other immune cells, such as B cells, natural killer cells, and dendritic cells, in the context of diabetic nephropathy.
  6. The article could benefit from a more in-depth discussion on the potential role of macrophages in the resolution of inflammation and tissue repair in diabetic kidney disease, as this information may guide the development of novel therapeutic strategies aimed at promoting the beneficial functions of these cells.
  7. The article does not provide a comprehensive overview of how current therapies for diabetic nephropathy, such as renin-angiotensin system inhibitors, glucose-lowering agents, and lifestyle interventions, may influence macrophage function and polarization.
  8. While the article highlights the need for further research to develop effective macrophage-targeted therapies for diabetic kidney disease, it does not provide a clear roadmap for future studies. The inclusion of specific recommendations for future research directions, such as identifying novel molecular targets, developing more specific macrophage modulators, and designing appropriate clinical trials, could enhance the impact of this review.

Reviewer 2 Report

Comments and Suggestions for Authors

In this really interesting review, Lin et al summarize the roles macrophages play in diabetic nephropathy in perpetuating inflammation through interactions with lymphocytes and other renal cells via extracellular vesicles, miRNA, and immune mediators. 

The review is thorough and well written.

The authors point out the roles of exosomes/EV in "Exosomes from tubular cells (HK-2) treated with high glucose and Epsin1 knockdown reduce macrophage activation, TNF-α, and IL-6 expression, and tubulointerstitial damage in C57BL/6 mice in vivo. " This is an interesting point. These exosomes contain a lot of miRNAs as discussed later in the review. Perhaps the authors can add another miR, miR-223: in RAW cells, it was shown that miR-223 influences their differentiation towards M1 or M2 phenotype by modulating NFKB and MCP-1. see PMID: 29778662

Perhaps some speculations on new treatments/drugs could be added at the end. How do the authors think that one can modulate macrophage transition to get a cure for diabetic nephropathy?
